# Personalizing chemotherapy drug selection using a novel transcriptomic chemogram

Kristi Lin-Rahardja [iD] [1,2]*, Jessica Scarborough [3], Jacob G. Scott [iD] [1,2,4]*

**1** Systems Biology and Bioinformatics, Case Western Reserve University School of Medicine, Cleveland, Ohio, United States of America, **2** Department of Genomic Medicine, Cleveland Clinic Research, Cleveland, Ohio, United States of America, **3** Internal Medicine, University of California San Francisco, San Francisco, California, United States of America, **4** Departments of Molecular Medicine, Physics and Biology, Case Western Reserve University, Cleveland, Ohio, United States of America

* kxl732@case.edu (KL-R); scottj10@ccf.org (JGS)

## Abstract

Gene signatures predictive of chemotherapeutic response have the potential to extend the reach of precision medicine by allowing oncologists to optimize treatment for individuals. Most published predictive signatures are only capable of predicting response for individual drugs, but most chemotherapy regimens utilize combinations of different agents. We propose a unified framework, called the chemogram, that uses predictive signatures to rank the relative predicted sensitivity of different drugs for individual tumors. Using this approach, providers could efficiently screen against many therapeutics to optimize chemotherapy at any time, whether it be for a treatment-naive tumor or a chemo-resistant tumor requiring a new treatment strategy. To demonstrate the utility of the chemogram, we used predictive signatures (extracted from a previously established method) in our framework to rank predicted sensitivity among drugs within cell lines. We then compared the rank order of predicted and observed response against each drug. Across most cancer types, chemogram-generated predictions were more accurate than predictions made by randomly generated gene signatures, signatures extracted from differential expression alone, and was comparable to another established method of drug response prediction. Our framework demonstrates the ability of transcriptomic signatures to not only predict chemotherapeutic response, but also correctly assign rankings of drug sensitivity on an individual basis. Additionally, scaling the chemogram to include more drugs does not compromise accuracy.

## Author summary

Standard chemotherapy typically takes a "one-size-fits-all" approach, even though individual patients may respond very differently to the same drugs. In our work, we present

**Data availability statement:** All data reported in the manuscript (GDSC and TCGA) are

publicly available. The code to download and clean data, perform the chemogram, and reproduce the figures as shown here are available on Github at github.com/kxlinr/evol-chemogram.

**Funding:** KLR and JGS were supported by a generous gift from the Carson Sarcoma Foundation.

a framework called the "chemogram" that would personalize chemotherapy decisions using only gene expression data from a patient's tumor. Our approach uses predictive gene signatures to rank how sensitive a tumor might be to different chemotherapy drugs and ranks predicted sensitivity among those drugs. This ranking could help doctors choose treatments that are more likely to work and avoid those that might cause side effects without real benefit to the patient. Importantly, our chemogram does not require complicated or time-consuming lab tests. Instead, it uses data that is relatively easy to collect and process. We tested the chemogram across hundreds of cancer cell lines and found it could accurately predict rank order of drug sensitivity in a few types of cancer. While further development and validation of the chemogram is necessary before it can be used in a clinical setting, we believe this approach has the potential to bring personalized chemotherapy within reach for many more patients, especially for those who don't qualify for targeted therapies.

## Introduction

The rapid proliferation and high variability of cancer cells enables tumors to quickly adapt to selection pressures, posing a significant challenge for individualized treatment. Most standard chemotherapy regimens adopt a one-size-fits-all approach, where patients will receive drugs that have been shown to be superior for a clinical trial cohort. Although such treatments may benefit a large portion of cancer patients depending on the cancer type, they are not guaranteed to be effective from individual to individual. One approach for addressing this inter-patient heterogeneity is precision medicine, particularly targeted therapy which takes advantage of certain actionable mutations that may be present in subsets of patients. However, most cancer patients lack such mutations and cannot benefit from this form of treatment [1]. 14% of all cancer patients are eligible for targeted therapy, and a mere 7% of all cancer patients respond to this form of treatment [2]. Thus, a majority of individuals with cancer will not receive precision medicine and instead are likely to be treated with a one-size-fits-all chemotherapy regimen.

Most cancer patients needing chemotherapy receive a combination of different drugs that can attack the tumor in multiple ways at once [3]. However, drug sensitivity is typically not measured for individuals prior to treatment. As a result, it is currently unclear if any given patient will truly benefit from every drug they receive, or if one or more of those agents is inducing negative side effects with limited benefit to the patient. Ideally, tumor biopsies would be screened for drug response prior to treatment so that only drugs with proven efficacy for the individual are administered. However, high-throughput drug screening requires substantial tissue, costly materials, and can take weeks to months depending on the facility. By the time the results are generated and returned to the clinician, the patient may have evolved a different sensitivity profile. For this reason, it is critical that pre-treatment analyses of drug response are performed quickly to allow timely implementation of treatment regimens.

An established clinical workflow is already in place to personalize drug regimens for patients who need antibiotics. These patients will often have an antibiogram performed on a sample of their infection, where a drug screen is performed on the sample to select antibiotics that effectively target bacteria without having an unnecessarily broad-spectrum treatment regimen [4]. This pipeline is effective in a clinical setting because bacteria grow far more rapidly than human tissue and antibiotics are less expensive and less toxic than chemotherapeutics. As aforementioned, drug screens for human tissue samples are time-consuming and

costly, making them impractical for clinical purposes [5]. To perform a drug screen, the tissue needs to be expanded *in vitro* so there are enough cells to test on. This further complicates matters because growing human tissue in a dish requires the cells to adapt to an incredibly different environment. If the tissue survives this selection process, the resulting cells are an imperfect model of the parent tumor. Alternatively, surgical samples can be seeded *in vitro* as 3-dimensional organoid cultures [6]. Organoid models of tumors do not require the same selection for adherent monoculture that typical 2D tissue culture requires, allowing tumor cells to expand in an environment more similar to where the sample was taken from. Despite this advantage, other challenges remain in culturing surgical samples as an organoid for the purpose of assessing drug sensitivity. Sample proliferation and outgrowth in an organoid culture is still not guaranteed, and materials for culturing the cells and assaying drug sensitivity can be expensive. Due to these significant hurdles, there are many ongoing efforts to predict drug response without requiring an *in vitro* drug screen.

Most published drug response prediction models utilize gene expression, whether it be exclusively or in conjunction with other data such as the chemical structures of drugs, genomics, or proteomics. As the availability of multi-omics data increases exponentially, more models are including such information to predict drug response [7–9]. However, clinical implementation of approaches that utilize multiple types of data may be slowed due to the cost and speed of obtaining all the information needed to employ the model. Model complexity can vary from relatively simple, such as gene signatures, to complex, such as neural networks and other machine learning methods [10]. Gene signatures are a straightforward tool that can be easily translated into a clinical setting, and several have already been incorporated into decision-making algorithms (e.g. Mammaprint [11,12], OncotypeDx [12,13], Prolaris [14,15], Decipher [16]). Previously, we established a method to extract gene signatures predictive of drug response based on exploiting convergent evolution by identifying gene expression patterns in genomically disparate tumors exhibiting sensitivity to the same agent [17]. Rather than predicting disease state and project outcome as the other aforementioned gene signatures have, our gene signatures are meant to guide treatment decisions by directly predicting chemotherapeutic response. In Scarborough et al., we derived a pan-cancer sensitivity signature for cisplatin across epithelial-origin cancer cell lines in The Genomics of Drug Sensitivity in Cancer (GDSC). We first identified similarities in differential gene expression across all epithelial-origin cancer cell lines that exhibited sensitivity to cisplatin, then compared their transcriptomes against cell lines that showed resistance to cisplatin. From this subset, the genes found to be the most highly co-expressed in tumor samples from The Cancer Genome Atlas (TCGA) were isolated to form the final cisplatin signature. We demonstrated that this signature can predict cisplatin response within cell lines from GDSC, and expression levels of the signature align with clinical trends observed in tumor samples from TCGA and the Total Cancer Care databases. As a preliminary validation, we conducted a case study in a novel muscle-invasive bladder cancer dataset to assess the signature's ability to estimate risk level and found the signature to be predictive among patients who have received cisplatin-containing treatment.

Utilizing this established signature model to generate gene signatures predictive of numerous common chemotherapeutics, we present a novel framework for predicting and ranking drug sensitivity among those therapies. Our proposed workflow, which we refer to as a transcriptomic chemogram, would serve a similar purpose as an antibiogram. Just as an antibiogram provides clinical decision support for antibiotic selection, a chemogram would inform oncologists which chemotherapies would be optimal for an individual cancer patient by using predictive gene signatures to assess sensitivity across multiple drugs. With this method, the sensitivity profile of a tumor can be extensively characterized in an efficient manner that

forgoes the need for a drug screen and can be performed at scale. Furthermore, a tumor's sensitivity profile could easily be reassessed with biopsies of recurrent tumors or metastatic lesions. A chemogram could also be used to determine appropriate treatments for patients who have cancer types that lack well-defined options for second-line chemotherapy. This is often the case for individuals with rare cancers that lack extensive research [18,19]. By using a chemogram to personalize chemotherapy and adjust treatment as the tumor evolves, patients with any type of cancer would be less likely to receive drugs that lack efficacy against their tumor at any time throughout treatment. We introduce this proof-of-concept and demonstrate the application of a transcriptomic chemogram *in vitro* using 10 predictive gene signatures.

## Results

### Gene signatures can predict and rank drug sensitivity of individual samples

Using the evolution-inspired extraction pipeline defined by Scarborough et al., we extracted 26 predictive gene signatures using the public datasets of The Cancer Genome Atlas (TCGA) and the Genomics of Drug Sensitivity in Cancer (GDSC) (see **Supplemental File 1**) [17]. For each drug for which we derive a signature for, this extraction method identifies differentially up-regulated genes between the most sensitive and most resistant GDSC cell lines across cancer types of epithelial origin. Three differential expression analysis methods are used, and only the genes found by all three methods are maintained for the next step. The resulting subset of genes is then filtered to maintain the most highly co-expressed genes among patient samples in TCGA. This method is repeated 5 times with a randomized 80% of cell lines, and the genes found in at least 3 of the 5 runs are included in the final signature.

For simplicity, we first demonstrated the chemogram with cisplatin, 5-fluorouracil, and gemcitabine, which are commonly used in various clinical settings. Later, we exhibit the scalability of the chemogram using the top 10 signatures by individual accuracy (Fig 1, see Methods). To do so, we use the method described by Scarborough et al. to assess drug sensitivity by generating a signature score (the median normalized expression among the genes in a given signature). For each tumor sample, one signature score can be generated for each predictive signature. The signature scores for each drug are then compared to each other to determine the rank order of predicted sensitivity, as depicted by Fig 2.

Using the gene expression data associated with each cell line provided by GDSC, we applied the chemogram to 616 untreated cell lines, 394 of which are of epithelial origin. First, raw gene expression values were normalized within samples by converting each value to a z-score. Normalization can also be performed by comparing to GAPDH expression, where each gene's expression value is divided by GAPDH expression for a given cell line. Either normalization method yields near-identical results downstream (S3 Fig). Using the normalized values, we then calculated the signature scores for each cell line-gene signature pair. The predicted sensitivity for the three drugs in each cell line was then ranked by descending signature score. The signature that yielded the highest score identified the drug predicted to cause the most cell death relative to the other two drugs. Likewise, the signature yielding the lowest score identified the drug predicted to cause the least cell death compared to the other drugs (Fig 2). Because each cell line's gene expression can vary, even within the same cancer type, each cell line can have a different prediction ranking among the three drugs. The signature scores for these cell lines are depicted in S4 Fig A. The ranking generated does not guarantee that the drug predicted to be the most effective will truly be effective, only that it will be

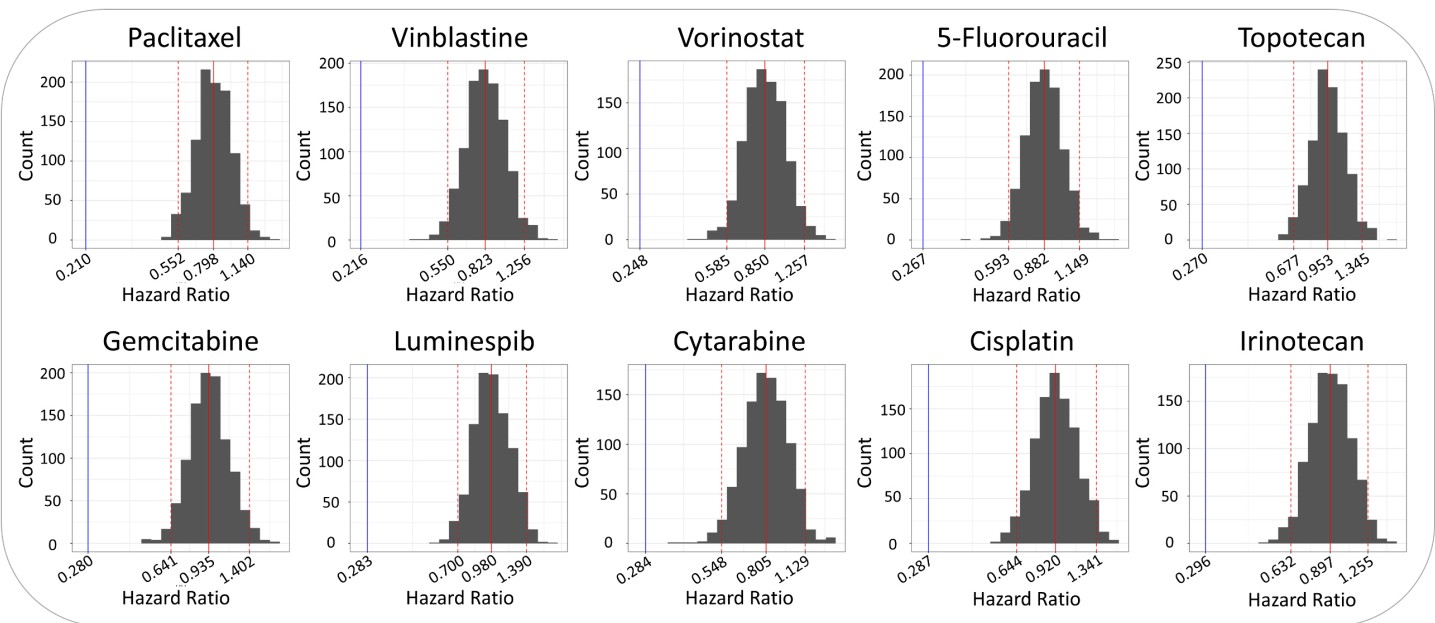

**Fig 1. Performance of all individual signatures exceeds the null distribution.** The performance of each signature was measured by calculating a hazard ratio using the cell line persistence curve method described by Scarborough et al., where GDSC cell lines are grouped by predicted sensitivity, and $IC_{50}$ is used in place of survival time. This generates results that resemble Kaplan-Meier data so that a hazard ratio can be calculated, but in this context is used to quantify *in vitro* survival differences. The hazard ratio for the extracted signature is denoted by the blue line. Hazard ratios were also calculated for 1000 randomly generated signatures of the same length, and these resulting values are depicted by the gray histogram. The dotted red lines mark the 95% confidence intervals for each null distribution, and the solid red line indicates the mean of the null signatures' hazard ratios.

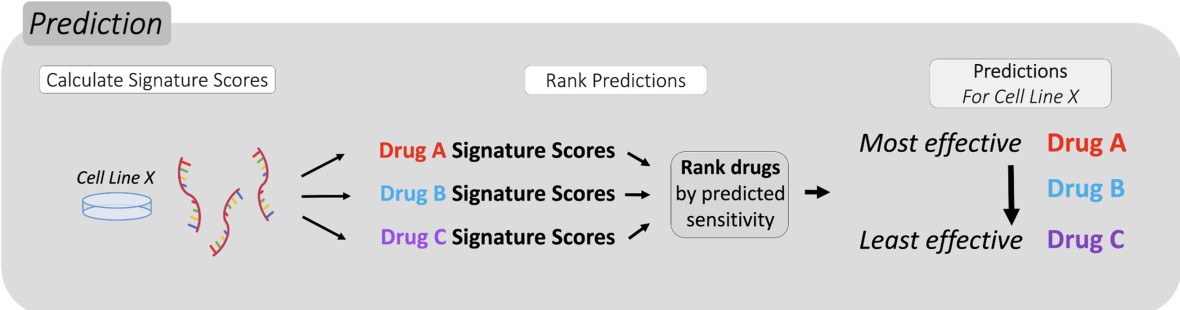

**Fig 2. Gene signatures predict the rank order of sensitivity for individual cell lines.** Signature scores for each drug were calculated to predict sensitivity in each cell line. Relative sensitivity rankings were determined by ordering the signature scores from highest to lowest value, where the highest value indicates the greatest predicted sensitivity.

relatively more effective than the other drugs included in the chemogram. Some drugs are frequently predicted to be more effective than others, and vice versa (Fig 4A).

## Prediction rankings are assessed for accuracy by comparing to observed survival against each drug

To assess the accuracy of the chemogram's predictions, we compared the predicted rank order of sensitivity (derived in Fig 2) to the rank order of observed survival against each of the three drugs (calculated in Fig 3). To measure observed survival in each cell line, we calculated

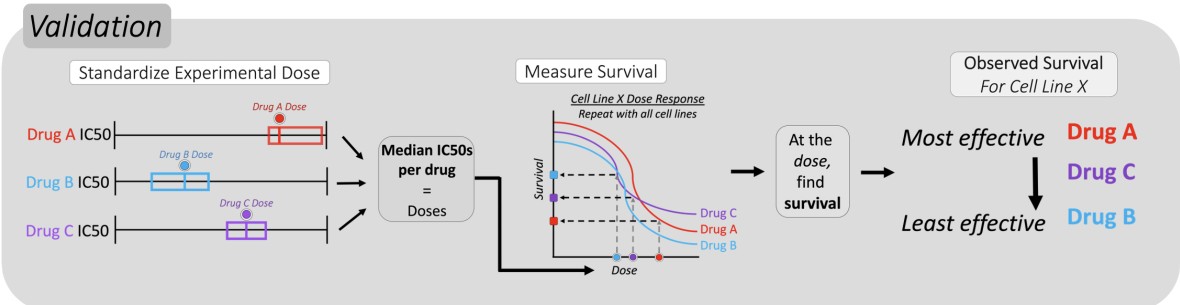

**Fig 3. Quantifying relative predictive accuracy of response signatures.** To determine the true sensitivity ranking, observed drug response was measured by calculating the surviving fraction of cells at a standardized dose for each cancer type. Standard doses were calculated by finding the median $EC_{50}$ of all cell lines in a cancer subtype.

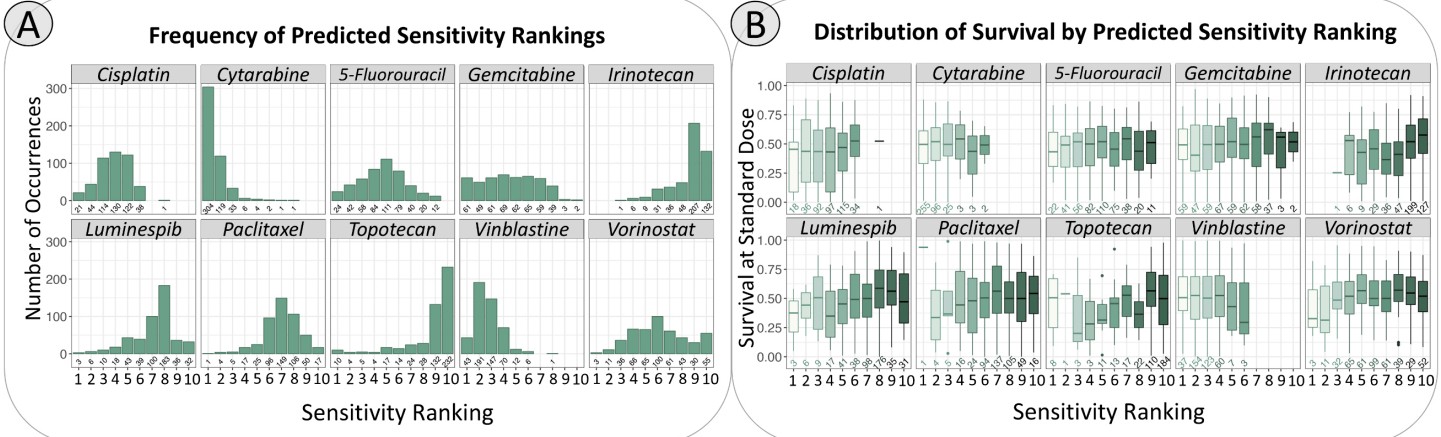

**Fig 4. Distributions of sensitivity ranking and survival per drug. A.** Histograms for each drug depict the frequency of a drug being placed in each of 10 possible sensitivity rankings. A rank of 1 indicates highest predicted sensitivity and rank 10 indicates lowest predicted sensitivity. Across epithelial-origin cell lines, cytarabine and vinblastine are often predicted to be more effective than the other drugs. Conversely, topotecan and irinotecan are often predicted to be less effective than the other drugs. The values marked underneath each bar specify the number of cell lines included in each ranking. **B.** Boxplots for each drug depict the distribution of survival at a standard dose. Within each drug's subpanel, distributions are stratified by sensitivity ranking. For instance, within the cisplatin panel, the leftmost boxplot shows the survival fractions against cisplatin for cell lines where cisplatin was predicted to be the most effective drug.

survival at a standard dose for each cancer type. We defined the standard dose per drug as the median $EC_{50}$ of all cell lines within a cancer type. For instance, to determine a standard dose for gemcitabine in all lung cancer cell lines, we find the median $EC_{50}$ against gemcitabine across all lung cancer cell lines. This is repeated for cisplatin and 5-fluorouracil. At these median $EC_{50}$s, we then calculated the fraction of surviving cells for individual cell lines using the raw dose-response data from GDSC fitted with the *gdscIC50* package [20].

Because cell lines of the same disease type can differ in their drug response, the measured survival can differ on an individual basis, similar to how different patients with the same type of cancer can have variable drug responses. This process was applied to all GDSC cell lines that had drug response data for all three drugs of interest, and the rank order of observed sensitivity was determined from this analysis. The fraction of surviving cells against the standardized doses for the three drugs across all of these cell lines is shown in S4 Fig B. Distribution of these survival values relative to sensitivity rankings per drug are shown in Fig 4B.

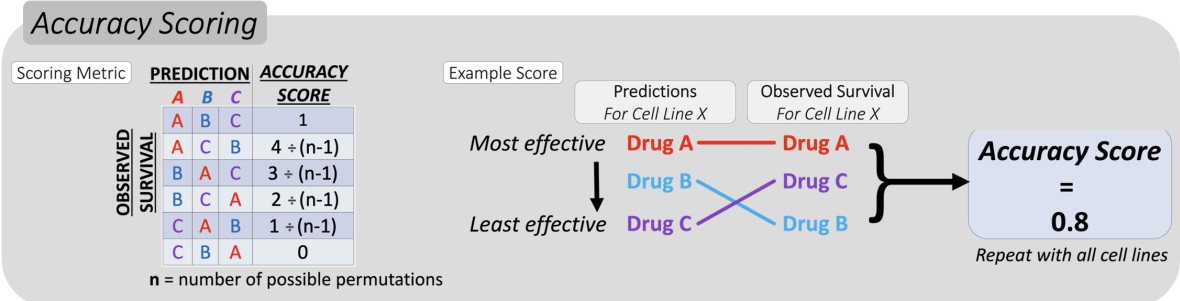

**Fig 5. Quantifying relative predictive accuracy of response signatures.** Accuracy was scored by comparing the predicted rank order of sensitivity to the observed rank order of sensitivity measured at a standard dose per cancer type. Accuracy is scored higher when a drug predicted to be more effective is shown to be so. This is done to more clearly reflect partially correct predictions.

## Using multiple signatures allows for the prediction of sensitivity rank order

To quantify the accuracy of the chemogram's predictions, the rank order of predicted sensitivity was compared with the rank order of observed survival per cell line. The accuracy of the predictions, relative to the observed survival, was scored from 0 to 1. With this scoring metric, 1 represents a perfectly accurate prediction of the survival rank order and 0 represents a prediction that was the opposite of the survival rank order. The remaining accuracy scores will be determined based on how close the drugs with greater predicted efficacy are to truly being more effective. The closer the drugs with greater predicted sensitivity are to truly showing more efficacy, the closer to 1 the accuracy score will be. Each increment of the accuracy score increases linearly, and the number of possible scores increases with the number of drugs used. An example of the scoring metric and its usage for 3 drugs is shown in Fig 5. The number of possible accuracy scores dramatically increases with the number of drugs included in the analysis. For further details on the scaling of the accuracy scoring system, the code used can be accessed through the GitHub repository. This method of scoring accuracy was preferred over other previously defined methods of comparing rank order (such as Kendall's Tau) for two primary reasons. Kendall's tau is more limited in the number of possible scores, making the resulting metric less descriptive. Using Kendall's tau to compare ranked lists of 3 drugs can only yield 4 possible values (number of possible outcomes = $((n(n-1))/2)+1$), whereas our method can produce 6 possible scores (number of possible outcomes = $n!$). When comparing ranked lists of 10 drugs, as we do later in the study, Kendall's tau can take only 46 distinct values, while our approach can yield 3,628,800. Our method is also more clinically relevant because we assign specific scores to every possible permutation of predicted order, allowing us to assign a higher score when drugs predicted to be more effective are actually observed to be. This could have important clinical implications if a clinician were to give these drugs sequentially based on the predicted ranking. For instance, following the example shown in Fig 5, if the predicted sensitivity ranking is drug A, B, then C, it would be more beneficial for the patient if the actual sensitivity ranking was drug B, A, then C, as opposed to B, C, then A. Other established statistical methods for comparing rank order would not be able to quantify these subtle distinctions.

We first demonstrated the chemogram using only three predictive signatures for cisplatin, gemcitabine, and 5-fluorouracil. Using these signatures, we performed the prediction, validation, and accuracy scoring as outlined in Figs 2, 3, and 5. Among the majority of cancer

cell lines in the GDSC dataset, the chemogram predicted sensitivity rank order with over 50% accuracy. Comparing these results to a null bootstrap, we found that the accuracy of the chemogram-generated predictions were consistently higher than the null distribution. To perform the null bootstrap, three randomly generated gene signatures (each the same length as one of the three evolutionary gene signatures) were used to predict and rank chemosensitivity in the same manner as before, as depicted by Fig 2. The accuracy of the random signatures' prediction would then be scored using the same method as described in Figs 3 and 5. This process was repeated for every cell line 1000 times, and the accuracy scores for each cell line in every iteration are represented by the blue boxplots in Fig 6A. The mean accuracy scores from each iteration of the null bootstrap, as well as the average scores from the chemogram, are shown in Fig 6B. Across all cancer types, the random signatures were only accurate about 50% of the time on average. The chemogram provides accurate predictions that exceed the null distribution in head and neck squamous cell carcinoma and prostate adenocarcinoma, but for other epithelial-origin cancers, it does not predict rank order of sensitivity better than random gene signatures. This result suggests that our framework does not have universal applicability and would be best utilized for specific disease sites.

## Signature-based chemograms are scalable for any number of drugs

To demonstrate the scalability of the chemogram, we repeated this analysis in its entirety with 10 signatures, all extracted using the same method as before [17]. The signatures used were associated with sensitivity to paclitaxel, vinblastine, vorinostat, 5-fluorouracil, topotecan, gemcitabine, luminespib, cytarabine, cisplatin, and irinotecan. The performance of each signature is represented in Fig 1. The process of prediction, validation, and scoring for accuracy is the same as depicted in Figs 2, 3, and 5, but with a greater number of drugs and associated signatures. Fewer cell lines were included in this analysis because several cell lines did not have associated drug response data for all 10 drugs. This portion of the analysis included 539 cell lines, whereas the 3-signature chemogram included 616. 351 of these cell lines were of epithelial origin for the 10-drug chemogram.

The accuracy of the chemogram was somewhat improved after it was scaled to use 10 drugs instead of 3. In most cancer types, the 10-signature chemogram consistently produced accuracy scores above 0.5, and the scores were distributed higher than that of the random predictions (Figs 6C and 6D). The null distributions were generated in the same way as described previously, but the bootstrap was performed with 10 random signatures rather than 3. The predictive accuracy of the 10-signature chemogram in head and neck squamous cell carcinoma is similar to that of the 3-signature chemogram, while the accuracy of the 10-signature chemogram in prostate adenocarcinoma is somewhat lower. The 10-signature chemogram accurately predicts the rank order of drug sensitivity in esophageal carcinoma and colorectal adenocarcinoma, yielding accuracy scores that are statistically significantly higher than those observed in the null distribution. This suggests that the chemogram's predictive utility may be enhanced with a broader drug set and could be particularly valuable in these two disease contexts.

## Comparison of other predictive models in a chemogram format

In addition to the null bootstrap, we compared the performance of other predictive models in a chemogram context, using these alternative methods to predict response among the same 10 drugs, across GDSC cell lines. We then ranked these individual predictions among drugs, compared them to the rank order of observed survival, and scored accuracy in the same manner outlined in Figs 3 and 5.

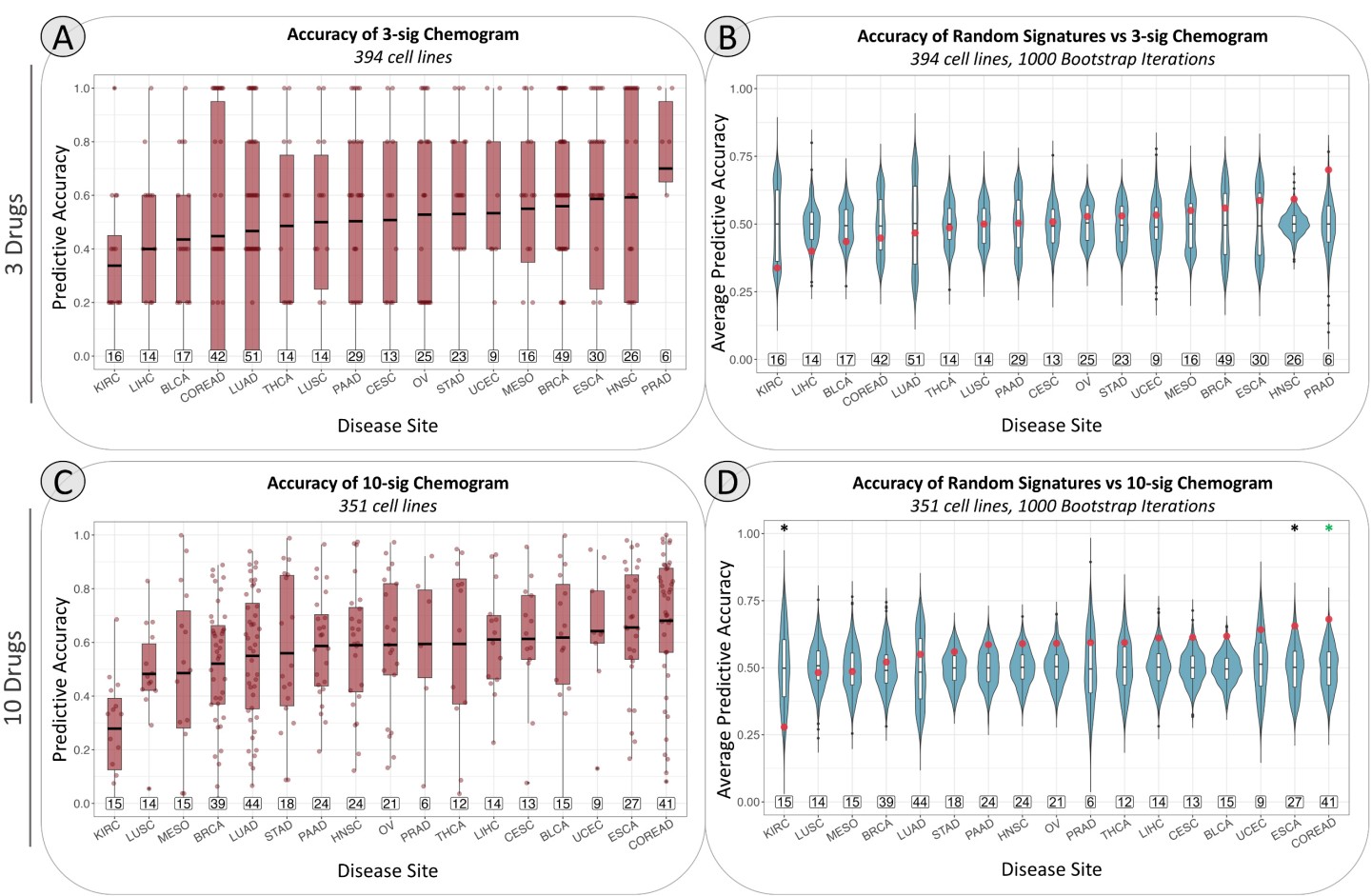

**Fig 6. Relative response predicted by the chemogram is more accurate than a null distribution.** Cancer subtypes are abbreviated using TCGA study abbreviations. The boxed numbers above the x-axis indicate the number of cell lines included in each cancer type. The lines in the center of the boxplots represent the mean accuracy, rather than the median. Asterisks indicate p ≤ 0.05 via unpaired Wilcoxon rank sum test. Green asterisks indicate FDR-adjusted p ≤ 0.05. In panels **A & C**, predictive accuracy scores for each cell line are shown, and each point represents the prediction accuracy in a single cell line. Each point in panels **B & D** reflects the mean accuracy score for all cell lines within a given cancer type for one iteration of the null bootstrap. Since the null bootstrap was iterated 1,000 times, each blue violin plot shows the distribution of 1,000 average scores. The red points correspond to the mean accuracy scores of the chemogram using our evolutionary signatures.

The first comparison we made was against signatures extracted from differential expression analysis using *limma* in GDSC cell lines, and maintaining the genes that were up-regulated in sensitive cell lines [21]. In contrast, the signatures used in our primary analysis involved 5-fold cross-validation, differential expression analysis with three different methods (including *limma*, *sam*, and *multtest*), and filtering through a co-expression network based on patient samples from TCGA [17]. In the same manner as for the original signatures, the simple signatures' predicted sensitivity per cell line was summarized by finding the median expression z-score of the signature genes. The accuracy of the predictions made with these simple signatures (Fig 7A in orange) was only slightly better than the randomly generated signatures.

The next model we tested was the *oncoPredict* package, which uses cell line gene expression, cell line drug response, and patient gene expression from the Cancer Therapeutics Response Portal (CTRP) and GDSC to predict cell line $IC_{50}$s [22]. We adapted the output of this model to rank predicted sensitivity among drugs by first applying the model to GDSC cell lines and predicting $IC_{50}$ for the 10 drugs of interest, then converting the $IC_{50}$s to z-scores

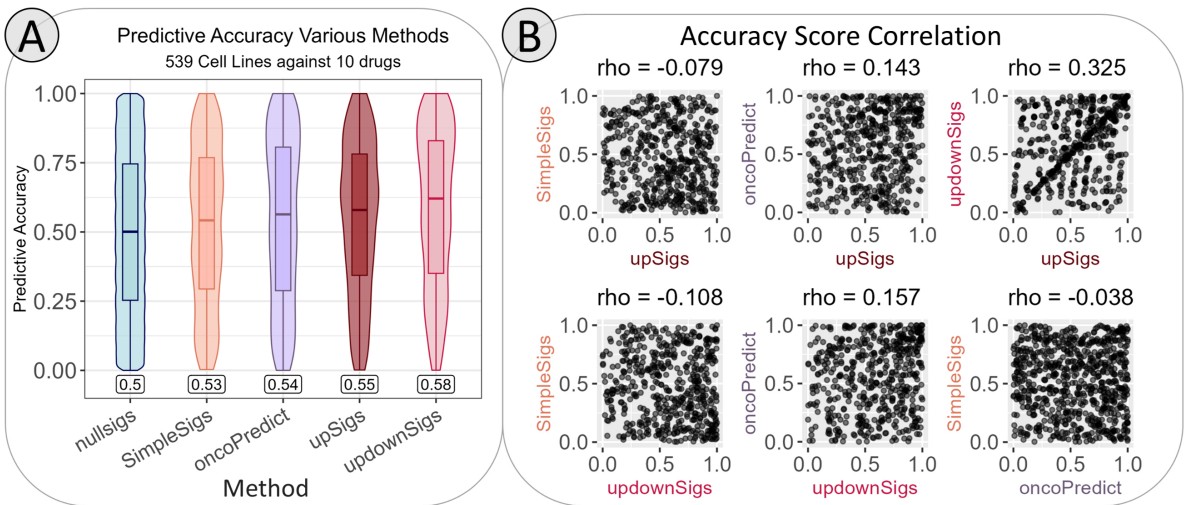

**Fig 7. Performance of various predictive models in a 10-drug chemogram setting. A.** *Predictive accuracy of different models used to rank predictive sensitivity among 10 chemotherapies.* As before, predictions were compared to observed survival at standardized doses and scored the same way as shown in Figs 3 and 5. The mean accuracy scores of each method are shown in the boxes above the x-axis. Using two-sided Wilcoxon rank-sum tests and a Bonferroni corrected significance threshold (alpha = 0.05/539 cell lines) to make all pairwise comparisons between methods, we found that updownSigs, upSigs (our original model), and oncoPredict were each significantly different from the null distribution. **B.** *Performance between models is generally not correlated.* Accuracy scores for each cell line are shown as individual points. Spearman's rho for each comparison is shown at the top of each plot. Most models are completely uncorrelated, while the original (up-regulated signatures only) and the up-/down-regulated signatures have a slight positive correlation. The lack of correlation between methods indicates that specific cancer subtypes do not drive overall performance improvements. This is also supported by S7 Fig.

per drug and across all cell lines. For each cell line, predicted sensitivity was then ranked from lowest z-score (most sensitive) to highest z-score (most resistant). The ranking was then scored as described previously, and the accuracy scores of all cell lines are shown in Fig 7A in the purple boxplot. Using *oncoPredict* and adapting its output to generate ranked sensitivity prediction yields very similar results to our original chemogram in terms of overall predictive accuracy, although its average accuracy score is somewhat lower than our method's.

The final alternative method we tested was using our original up-regulated sensitivity signatures along with down-regulated sensitivity signatures that were also extracted using the Scarborough et al. pipeline [17]. Originally, we focused on demonstrating our method only using genes that were up-regulated in sensitive samples because gene up-regulation, as opposed to down-regulation, is more reliably detectable and tends to be more statistically robust [23,24]. In this approach, we summarized predicted drug sensitivities by subtracting the median expression z-score of the down-regulated gene signature from the median expression z-score of the up-regulated signature. Overall, the accuracy of this approach was noticeably higher than that of our original approach using only up-regulated gene signatures, but the difference was not statistically significant (Fig 7A in pink). Additionally, the accuracy scores for each cell line between this method and the original were only somewhat correlated (Spearman's rho = 0.325, Fig 7B), indicating that this alternative method was not consistently better than up-regulated signatures alone. In observing these results stratified by cancer types, it is evident that using both up- and down-regulated signatures can be more effective in certain cancer types than others, but not necessarily all cancer types (S6 Fig and S7 Fig). Moving forward, different combinations of signature types (up-regulated only, or combining up- and down-regulated) could be optimized on a cancer type-specific basis.

In the future, *in vitro* validation of our method can be accomplished using patient-derived xenografts and surgical samples, preferably with esophageal carcinoma or colorectal adenocarcinoma samples, where the chemogram was shown to have significant predictive power. This would be done by measuring gene expression of the samples and applying the chemogram, and then assaying the samples' dose response against each of the drugs included in the chemogram. To find the rank order of observed sensitivity that we can compare the predictions against, we can observe the fraction of surviving cells at the same standardized doses we calculated in this study, where we used the median $EC_{50}$ across cells within the same disease type.

## Discussion

Standard chemotherapy generally takes a one-size-fits-all approach, where patients without targetable mutations will receive some combination of drugs that work for most patients, or improve outcomes to some degree compared to previous regimens, based on clinical trials. However, every cancer patient has a unique chemosensitivity profile, and what may be optimal for a population is not guaranteed to work for an individual. Further, increased benefits from combination chemotherapy may largely be attributed to at least one, but not necessarily all, of the drugs being effective [3,25–27]. In this scenario, patients are receiving more than what is necessary for effective treatment and experiencing excessive toxicity as a result. Precision medicine, particularly targeted therapy, improves on the issues imposed by the one-size-fits-all approach to chemotherapy and interpatient tumor heterogeneity by targeting driver mutations present in a tumor, and such therapies are only available to patients who harbor such mutations. However, few patients are eligible for targeted therapy and efficacy can be limited even for those who harbor actionable mutations [2]. Even patients who respond to treatment initially are not immune to disease recurrence due to the evolution of resistance [28,29]. Personalizing chemotherapy using predictive gene signatures would make precision medicine much more accessible to patients, regardless of mutation status, while also aiding the treatment of patients with chemoresistant tumors.

The chemogram can be easily adapted for any number of drugs that a predictive signature exists for and does not require a plethora of time-consuming assays to be performed. While this framework is still in its early stages, future implementation of this method would be similar to that of an antibiogram, which is used to determine which antibiotics a patient is sensitive to for maximized treatment efficacy [4]. To use a chemogram, gene expression of a tumor biopsy would be measured, predictive signatures would be compared against the gene expression profile, sensitivity scores would be calculated, and the best drugs to give a patient can be determined. Once the results are generated, clinicians would receive a ranked list of drugs, avoiding confusion and leaving little room for subjective interpretation. Since the chemogram is based on gene expression, its application is not limited to patients with actionable mutations the way targeted therapy is. Because of this, the chemogram has the potential to help many more patients via treatment personalization.

The chemogram would be an ideal tool for guiding evolution-informed therapy. Evolution-informed therapy is an approach to treatment where a drug regimen is continuously adjusted according to the phenotypic changes in the disease. This method has been explored in theoretical models and bacteria, but has yet to be tested in the context of cancer [30–33]. Tumors are constantly evolving and adapting to selective pressures, so this dynamic approach to treatment would be highly relevant for use with chemotherapeutics. The simplicity and flexibility of the chemogram makes it a promising device to guide an evolution-informed chemotherapy regimen. Depending on the physical accessibility of the tumor, regular biopsies could be taken

and assessed using the chemogram to adjust treatment as a tumor continues to evolve so that the patient is continuously treated as effectively as possible. Alternatively, the chemogram and the signatures used with it might be adapted for use with liquid biopsies and measure circulating tumor DNA.

Across cancer types, clinicians usually treat patients with combination therapy because these regimens generally yield more favorable responses compared to single agents [3]. Several hypotheses for this benefit exist. Traditionally, drug combinations have been assembled based on the idea that the probability of resistance arising against multiple drugs with distinct mechanisms at once is less probable than resistance arising against a single agent [34]. Interactions among drugs in a combination can also exist, and there are multiple models for these interactions. One is the Bliss independence model, where drugs in a combination act independently of each other, and the combined effect of the combination is equal to the sum of its parts [35]. Similarly, the Loewe model of drug additivity assumes that the effect of a drug combination is equal to the sum of its components, but this model does not require the assumption that each drug acts independently of one another [36]. From these models, synergistic and antagonistic interactions can also be defined, where the effect of one drug enhances or impedes the effect of another drug, respectively. Synergistic combinations produce a therapeutic effect greater than the sum of its parts, while antagonistic combinations produce an effect less than the sum of its parts. Retrospective studies on clinical trial data have revealed that the benefit of using combination chemotherapy is conferred primarily by the independent drug action model, where one treatment does not change the activity of another [25,26]. Thus, treating a tumor with multiple drugs increases the probability that a patient will respond to at least one of the drugs being used [27,37]. This means that if a patient is resistant to one or more drugs in the combination, this approach risks increased toxicity with limited clinical benefit. With this understanding, patients should ideally receive drugs with known efficacy against their tumor, minimizing unnecessary toxicity from agents that are ineffective against their tumor [38].

The chemogram, as presented in our study, was limited to monotherapy because the predictive signatures were only extracted for one drug at a time. To better synergize the chemogram with standard practices, signatures that are predictive of drug combinations could be included in the screen. For instance, a chemogram could include a signature for drug A, drug B, and drugs A and B together. This would help clinicians determine if all drugs in a standard combination are appropriate for use on a case-by-case basis, thereby preventing excessive toxicity. Alternatively, potential drug interactions among different combinations could be assessed after the ranked list is generated. This might be accounted for by calculating an interaction score of some kind among the drugs predicted to be most effective. For example, if drugs A, B, and C are the top 3 drugs, one could determine whether the combination is appropriate by calculating a score for the combinations AB, AC, BC, and ABC, then treating the patient with the combination that yields the highest synergy score. Several ongoing efforts are being made to create a method for predicting and quantifying drug interactions, but many of them are based on different definitions of synergy and do not agree with each other as a result [39–41]. However, the potential problem of adverse drug interactions in patients may not be a serious concern based on recent findings. A vast majority of approved drug combinations only seem to exhibit additivity—true synergy appears to be rare, as is antagonism [3,25]. However, rarity does not eliminate all potential, and drug interactions should be kept in mind as more research continues to be done on drug interactions at a patient level.

One hurdle for future clinical implementation of the chemogram is the need to define a sensitivity score threshold for individual signatures that is relevant across patients. With the current approach, the ranked list of drugs generated by the chemogram is relative to each

individual patient. For instance, if a patient is resistant to all of the drugs screened in the chemogram, their caregiver would still receive a ranked list even if the top-ranked drugs are unlikely to be effective. Defining a threshold for individual sensitivity scores would allow clinicians to make clear distinctions about what drugs would be effective for treatment and should be considered a valid option for treatment. If many drugs exceed the threshold, the chemogram would aid in determining the best options among those choices. Signatures that score below the threshold would indicate which drugs might do little to reduce tumor burden and should be avoided. Given that different agents will have different overall clinical efficacy, the probability distributions of efficacy per cancer type could be included in the calculation of a sensitivity score threshold. Since our demonstration of the chemogram was done using cell line data, we could not define such a threshold. A dataset similar to The Cancer Genome Atlas (TCGA) that contains gene expression data for thousands of tumors would be optimal, but TCGA itself does not contain treatment history. Furthermore, finding a well-annotated clinical dataset that is racially diverse is also incredibly important for building any robust clinical decision-making tool. GDSC, which we used to extract the signatures and demonstrate the chemogram, does not report information on the race of the individuals whose cells were derived to make the cell lines included in the repository. The Cancer Genome Atlas, a large clinical dataset that we used alongside GDSC to extract the signatures, consists of data predominantly gathered from Caucasian patients (73.3%) [42]. Numerous studies have found differences in drug response among racial/ethnic groups [43–46]. If most biomarkers are extracted from datasets that are not as racially diverse as the populations they will be applied to, the applicability of such biomarkers would likely be much lower than pre-clinical studies may indicate. Any further extrapolation or validation of this work should be more racially inclusive so that the chemogram is robust and effective for all cancer patients.

Although the chemogram we present here is still in its early stages of development, there is much promise it holds for the future of treatment planning. The use of evolution-inspired gene signatures allows the framework to be accessible for patients with and without targetable mutations, thereby greatly extending the reach of precision medicine. Because the chemogram can be used multiple times throughout treatment, it can also be an effective aid for combating drug resistance. The simplicity of our method makes clinical translation quite straightforward to perform at scale and easy for clinicians to interpret. Ultimately, chemotherapy could be effectively optimized for every cancer patient.

## Methods

### Data collection and pre-processing

All data cleaning, analysis, and plotting were performed using R (Version 4.3.0) with RStudio [47]. Associated code for the following subsections can be found on Github (see **Data & Code Availability**) under the Scripts/setup.Rmd.

**GDSC gene expression data.**  Microarray mRNA expression data for 1018 cell lines was downloaded from GDSC, which can be accessed from https://www.cancerrxgene.org/www.cancerrxgene.org/ [48]. The expression data was collected using the Human Genome U219 96-Array Plate with the Gene Titan MC instrument (Affymetrix). This data was normalized using the robust multi-array analysis (RMA) algorithm [49,50]. For signature extraction, expression data was then converted to z-scores per gene within samples. The chemogram was used with both z-score normalized expression as well as GAPDH-normalized expression, and the results for each were nearly identical (S3 Fig B). GAPDH was selected as the housekeeping gene to normalize the data to because of its consistently high expression level among all cell lines (S3 Fig A). Cancer cell lines of

epithelial origin were defined based on these GDSC tissue descriptors: head_and_neck, oesophagus, breast, biliary_tract, large_intestine, liver, adrenal_gland, stomach, kidney, lung_NSCLC_adenocarcinoma, lung_NSCLC_squamous-_cell_carcinoma, mesothelioma, pancreas, skin_other, thyroid, Bladder, cervix, endometrium, ovary, prostate, testis, urogenital_system_other, uterus.

**GDSC drug response data.** Both raw drug response data and reported $IC_{50}$ values for 809 cell lines were downloaded from GDSC. Reported $IC_{50}$s were used for signature extraction and raw drug response data was used for chemogram validation presented by Fig 3. The raw dose-response data was cleaned and normalized using the *gdscIC50* package in R [20]. Normalization involved converting raw fluorescence values to percent viability, relative to the negative and positive controls. Dose-response curves for each cell line-drug pair were fitted using a non-linear mixed effects model with the *fitModelNlmeData* function defined in the *gdscIC50* package. The fitted equations generated from this were used to calculate $EC_{50}$s and observed survivals per cell line.

Not all cell lines had associated drug responses for all of the drugs used in our demonstration, which reduced the number of cell lines we could include in our analysis. Excluding cell lines with an unclassified cancer subtype, we were left with 763 cell lines for the 3-signature chemogram and 546 cell lines for the 10-signature chemogram. From these respective cohorts, there were 398 epithelial-origin cell lines for the 3-signature chemogram and 356 epithelial-origin cell lines for the 10-signature chemogram.

**TCGA gene expression data.** The Cancer Genome Atlas (TCGA) gene expression data was used for signature extraction. The RNA-Seq by Expectation Maximization (RSEM) normalized gene expression was downloaded using the *RTCGAToolbox* package (version 2.30.0) [51]. Expression values were measured via Illumina HiSeq RNAseq V2 and were log2 transformed.

## Sensitivity signature extraction

Predictive signatures were extracted using the pipeline described by Scarborough et al. [17]. This work made use of the High Performance Computing Resource in the Core Facility for Advanced Research Computing at Case Western Reserve University. The GDSC data was partitioned into 5 folds, where each fold contained a randomized 80% of all cell lines in the dataset. For each signature, GDSC cell lines (within each fold) were ordered from most resistant to most sensitive. Cell lines in the top 20-30% (20% for all signatures except 5-fluorouracil, which used 30%) of each extreme were identified, and differential expression was measured between the groups using limma, sam, and multtest [21,52,53]. The genes found to be differentially expressed by all 3 methods were used as seed genes for a co-expression network that was built using TCGA expression data. Seed genes found to be highly co-expressed in at least 3 of the 5 folds were included in the final signature.

We sought to extract predictive signatures for many commonly used cancer drugs, but ultimately narrowed down the number of signatures to 10 for demonstration in this study. Firstly, many of the drugs in GDSC were not able to produce gene signatures using our pipeline. We filtered our signature list to only include those that predicted response to chemotherapies, leaving us with 26 signatures. For these remaining 26 signatures, we assessed the accuracy of each using the hazard ratio metric described by Scarborough et al. (see **Supplemental File 1**). The performance of each signature was assessed by calculating the signature score (median z-score expression of the signature genes) for all GDSC cell lines, then comparing the distribution of $IC_{50}$ between cell lines in the highest and lowest quintiles of the signature scores.

From this, we then calculated a hazard ratio, and this metric was used to rank all 26 signatures from best to worst accuracy. The top 10 performing signatures were used in this study. The hazard ratio of these 10 signatures compared to a null distribution is shown in Fig 1. The null distribution was generated by using 1000 randomly generated gene signatures to stratify cell lines and calculate hazard ratios (per drug, all 1000 random signatures were the same length as the original signature). Based on the hazard ratio, we selected the top 10 signatures to demonstrate the chemogram. All but the irinotecan signature outperformed the cisplatin signature, which is the same as CisSig derived by Scarborough et al. All the signatures were extracted using the same parameters as the cisplatin signature, with the exception of the irinotecan, topotecan, and vorinostat signatures, which used a differential expression cutoff of 0.15 instead of 0.2.

## Statistics

All statistics were performed in R. Wilcoxon rank sum tests were run for the data shown in Fig 6 and S5 Fig.

## Supporting information

**S1 File. Full list of genes included per drug.**
(XLSX)

**S2 File. Full table of signature genes.**
(PDF)

**S1 Fig. Genes included per signature.** Genes that appear in more than one of the 10 signatures used are listed. Out of 160 unique genes across 10 signatures, 44 are present in multiple signatures. For the full table including genes that appear in only 1 signature, see **Supplemental File 2**. To see entire signatures listed out per drug, see **Supplemental File 1**.
(TIFF)

**S2 Fig. Gene expression normalization. A.** Among all cell lines in GDSC, GAPDH consistently exhibits high expression levels. As such, gene expression (for use within the chemogram) can be normalized to GAPDH expression within each cell line rather than normalizing by converting expression values to z-scores. **B.** Signature scores derived for all drugs and cell lines are compared when calculated using GAPDH-normalized expression or expression z-scores. Signature scores derived from either normalization method are highly correlated for all signatures. Spearman's rho is indicated at the top of each subpanel.
(TIFF)

**S3 Fig. Distributions of IC$_{50}$s per drug.** The natural log of IC$_{50}$s calculated using the *gdscIC50* package are shown in the histograms. Vertical black lines denote the standard doses used for validation as described in Fig 3.
(TIFF)

**S4 Fig. Heatmaps of signature scores and survivals at standard doses per cell line. A.** Signature scores for all cell lines are calculated for all 10 gene signatures. Each column depicts the signature scores generated from one signature, and each row represents a single cell line. The higher the score, the greater the predicted sensitivity. Signature scores across cell lines tend to be similar within drugs. **B.** The fraction of surviving cells was measured at standardized doses against each of the 10 drugs. Each column represents survival against one drug, and each row represents a single cell line in the same order as shown in panel B. A higher

survival indicates greater resistance against a drug, and a lower survival indicates greater sensitivity against a drug.
(TIFF)

**S5 Fig. Relative response predicted by the chemogram across all cancer types.** This figure is formatted the same as Fig 6, but includes both epithelial and non-epithelial cancer cell lines. Cancer subtypes are abbreviated using TCGA study abbreviations. The boxed numbers above the x-axis indicate the number of cell lines included in each cancer type. The lines in the center of the boxplots represent the mean accuracy, rather than the median. Asterisks indicate $p \leq 0.05$ via unpaired Wilcoxon rank sum test. Green asterisks indicate FDR-adjusted $p \leq 0.05$. The results are very similar to that of only the epithelial cancers for both the 3-drug and 10-drug chemograms.
(TIFF)

**S6 Fig. Accuracy scores by cancer type between up-regulated signatures only or up-and down-regulated signatures in a chemogram setting.** Cancer subtypes are abbreviated using TCGA study abbreviations. Both epithelial-origin and non-epithelial-origin cancers are included. The boxed numbers above the x-axis indicate the number of cell lines included in each cancer type. The lines in the center of the boxplots represent the mean accuracy, rather than the median. Each point represents the predictive accuracy scores for a cell line. The red boxplots correspond to the accuracy scores of the chemogram using our up-regulated signatures only, while the pink boxplots correspond to the accuracy scores of the chemogram using both up- and down-regulated signatures. With $\alpha = 0.05$, only skin cutaneous melanoma (SKCM) had significant differences between method performance based on a two-tailed Wilcoxon rank-sum test with either a Bonferroni-corrected threshold or with an FDR-corrected p-value.
(TIFF)

**S7 Fig. Correlation of accuracy scores between methods by cancer disease site.** The same plot as Fig 7B is shown here, but with each point colored by the cancer subtype associated with the cell line being presented. No major trends are apparent among any particular cancer subtype, although accuracy scores for SKCM from the up- and down-regulated signatures together are noticeably higher than the up-regulated signatures alone.
(TIFF)

**S8 Fig. Statistical differences between methods.** Pairwise comparisons among all methods per cell line were made using two-tailed Wilcoxon rank-sum tests. P-values for each comparison are shown as individual points, with color corresponding to the comparison being made. Log10(p-value) is shown on the x-axis of the **left panel**, while log10(false discovery rate-adjusted p-value) is shown on the x-axis of the **right panel**. The y-axis denotes cancer type, and the boxed number to the right of the y-axis shows the number of cell lines included in the associated cancer type. In the **left panel**, Bonferroni-corrected significance thresholds are marked with a vertical black line at $\alpha = 0.05$ divided by the number of cell lines in the cohort. In the **right panel**, a black line marks the significance threshold of $\alpha = 0.05$.
(TIFF)

## Author contributions

**Conceptualization:** Jacob Scott.

**Data curation:** Kristi Lin-Rahardja, Jessica Scarborough.

**Formal analysis:** Kristi Lin-Rahardja, Jessica Scarborough.

**Investigation:** Kristi Lin-Rahardja.

**Methodology:** Kristi Lin-Rahardja, Jacob Scott.

**Project administration:** Jacob Scott.

**Supervision:** Jessica Scarborough, Jacob Scott.

**Visualization:** Kristi Lin-Rahardja.

**Writing - original draft:** Kristi Lin-Rahardja.

**Writing - review & editing:** Kristi Lin-Rahardja, Jessica Scarborough, Jacob Scott.

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
