## [Decision Letter · Decision Letter 0]

5 Jun 2025

PCOMPBIOL-D-25-00615

Personalizing chemotherapy drug selection using a novel transcriptomic chemogram

PLOS Computational Biology

Dear Dr. Scott,

Thank you for submitting your manuscript to PLOS Computational Biology. After careful consideration, we feel that it has merit but does not fully meet PLOS Computational Biology's publication criteria as it currently stands. Therefore, we invite you to submit a revised version of the manuscript that addresses the points raised during the review process. Please note that the revisions demanded by reviewer 1 under "Major comments" are not optional.

Please submit your revised manuscript within 60 days Aug 05 2025 11:59PM. If you will need more time than this to complete your revisions, please reply to this message or contact the journal office at ploscompbiol@plos.org. Please include the following items when submitting your revised manuscript:

We look forward to receiving your revised manuscript.

Kind regards,

Marc Robinson-Rechavi

Academic Editor

PLOS Computational Biology

Ilya Ioshikhes

Section Editor

PLOS Computational Biology

**Additional Editor Comments:**

I agree with Reviewer 1 that you need to compare not just to random signatures, but to state of the art predictions. This point is critical. The reviewer makes excellent suggestions, which I invite you to follow.

**Journal Requirements:**

At this stage, the following Authors/Authors require contributions: Kristi Lin-Rahardja, and Jessica Scarborough. Please ensure that the full contributions of each author are acknowledged in the "Add/Edit/Remove Authors" section of our submission form.

5) We have noticed that you have uploaded Supporting Information files, but you have not included a complete list of legends. Please add a full list of legends for your Supporting Information files after the references list.

6) We notice that your supplementary Figures are included in the manuscript file. Please remove them and upload them with the file type 'Supporting Information'. Please ensure that each Supporting Information file has a legend listed in the manuscript after the references list.

7) Some material included in your submission may be copyrighted. According to PLOSu2019s copyright policy, authors who use figures or other material (e.g., graphics, clipart, maps) from another author or copyright holder must demonstrate or obtain permission to publish this material under the Creative Commons Attribution 4.0 International (CC BY 4.0) License used by PLOS journals. Please closely review the details of PLOSu2019s copyright requirements here: PLOS Licenses and Copyright. If you need to request permissions from a copyright holder, you may use PLOS's Copyright Content Permission form.

Potential Copyright Issues:

i) Figure 2. Please confirm whether you drew the images / clip-art within the figure panels by hand. If you did not draw the images, please provide (a) a link to the source of the images or icons and their license / terms of use; or (b) written permission from the copyright holder to publish the images or icons under our CC BY 4.0 license. Alternatively, you may replace the images with open source alternatives. See these open source resources you may use to replace images / clip-art:

8) Please send a completed 'Competing Interests' statement, including any COIs declared by your co-authors. If you have no competing interests to declare, please state "The authors have declared that no competing interests exist". Otherwise please declare all competing interests beginning with the statement "I have read the journal's policy and the authors of this manuscript have the following competing interests:"

**Reviewers' comments:**

Reviewer's Responses to Questions

**Comments to the Authors:**

Reviewer #1: The review is uploaded as an attachment.

Reviewer #2: Lin-Rahardja and colleagues build upon an important recent study which described an algorithm for predicting drug sensitivity of human cancer cell lines using publicly available gene expression and drug sensitivity data from GDSC and CTRP (Scarborough et al, 2023, NPJ Precision Oncology). That prior work addressed an important unmet need to identify which patients are most likely to benefit from a particular chemotherapy, with the first proof of concept focused on cisplatin. This new article applies the previously described algorithm to 9 other cancer chemotherapies, nearly all of which are widely used in clinical oncology, and all of which lack clear biomarkers of response. Testing their method’s applicability to this substantially expanded set of clinically significant chemotherapies is an important goal, especially given the lack of existing predictive biomarkers for these therapies. The introduction persuasively describes the need for greater precision in the choice of chemotherapies, and how a gene expression-based algorithm, if effective, has greater potential for widespread use than in vitro drug screening on biopsied material.

The study develops tissue-agnostic ‘chemograms’ based on publicly available data. 5-fold cross-validation is a suitably rigorous means to classify resistant versus sensitive cell lines. Given the lack of consensus methods for calling differentially expressed genes, the authors apply multiple algorithms and select genes found in the intersection of all methods, which I anticipate should add to the robustness of results. Predictive performance is assessed on a tissue-specific basis, which shows that 3-gene and 10-gene chemograms outperform random signatures for some tissue types and for some drugs. This is to be expected, and it would be unreasonable to expect the approach to be effective for all tissue types and all drugs (I would be skeptical if a study made such a claim).

Defining the specific contexts where chemograms could be effective is the most important result of the study. As written, the results and conclusion sections emphasize the overall evidence for the predictive potential of chemograms. I think one or both sections would benefit from more specific statements about which drug classes and tissue types appear amenable to this approach, recognizing that it would be unreasonable to expect universal applicability, and indeed the evidence does not suggest universal applicability.

Distributions of IC50s in predicted sensitive and resistant cultures are compared by Hazard Ratio. As this is a non-standard use of the Proportional Hazards model, it would be helpful for the methods or discussion to concisely mention that these hazard ratios quantify in vitro differences and do not indicate or predict time-to-event hazard ratios in humans. The graphs of IC50 distributions (presented like Kaplan-Meier plots) for each drug would be a valuable inclusion in the supplemental materials. Such a graph was shown in Scarborough 2023 in Figure 3C for cisplatin; the same should be shown here for each of the other 9 chemotherapies.

One last point would be a valuable addition to the discussion, regarding the comparison of many drugs, which was not relevant to this group’s 2023 article on cisplatin only. Given some set of drugs to choose between for some cancer type, in general these drugs will not have equal overall clinical efficacy; some drugs will be more effective than others on average. Consider for example 2 drugs, where one has a 50% response rate, and the other has a 20% response rate. If a patient is predicted to have ‘median’ sensitivity to the first drug, but ‘top 30%’ sensitivity to the second drug, then rank order may predict the second drug to be better, but when considering differences among drugs in overall response rates, such a patient would be better off receiving the first drug with the higher response rate. This complication needn’t be solved in this pre-clinical proof-of-concept study, but its relevance to future clinical applications of choosing between drugs deserves discussion.

**Have the authors made all data and (if applicable) computational code underlying the findings in their manuscript fully available?**

Reviewer #1: Yes

Reviewer #2: Yes

PLOS authors have the option to publish the peer review history of their article (what does this mean?). If published, this will include your full peer review and any attached files.

Reviewer #1: No

Reviewer #2: No

**Figure resubmission:**
---

## [Decision Letter · Decision Letter 1]

11 Aug 2025

Dear Dr. Scott,

We are pleased to inform you that your manuscript 'Personalizing chemotherapy drug selection using a novel transcriptomic chemogram' has been provisionally accepted for publication in PLOS Computational Biology.

Best regards,

Marc Robinson-Rechavi

Academic Editor

PLOS Computational Biology

Ilya Ioshikhes

Section Editor

PLOS Computational Biology

Reviewer's Responses to Questions

**Comments to the Authors:**

Reviewer #1: Below comments address only the Revision, original review is available in previous submission.

The authors did a good job addressing the main original point about the need to compare with another methods. They did compare to two types of other methods, namely one based on simpler approach for deriving differentially expressed genes and one based on simple machine learning approach. Comparison with more recent, deep learning based state-of-the art models for drug sensitivity prediction would still enrich the paper, but the performed comparison still addresses the original comment.

The authors did address the comment about including also down-regulated genes, creating another variant of their method which yielded better performance when looking globally at all cell lines. This yielded interesting results when looking at per-cancer-type basis as in Figure S7, which might enrich the discussion.

I understand the choice of 10 drugs but still I am not entirely convinced about using 3-drugs set as an example. Also, does the fact that Chemogram can be extracted for only 26 drugs is not a significant limitation of the method? I believe that it’s not discussed with more emphasis as currently written.

I appreciate clarifications about scoring metrics.

Reviewer #2: The revised manuscript fully addresses the comments of my initial review, which were chiefly about some further explanations of methods and disease settings in which the current data suggest the greatest potential for applicability. The revised article accomplishes these tasks very nicely. The revised discussion especially makes a compelling case for the clinical need and the potential impact of chemograms and related approaches for precision oncology, which stand out as a very distinct approach, of use for different drugs, when compared with genome sequence-based inhibition of oncogenes. It is true, as hinted in comments by reviewer 1 also, that improved predictive accuracy remains desirable, but the breadth of application of this study and its inclusion of complete source data and software for reproducibility makes this paper a valuable contribution to progress on the important and tough challenge of precision cancer medicine.

**Have the authors made all data and (if applicable) computational code underlying the findings in their manuscript fully available?**

Reviewer #1: Yes

Reviewer #2: Yes

PLOS authors have the option to publish the peer review history of their article (what does this mean?). If published, this will include your full peer review and any attached files.

Reviewer #1: No

Reviewer #2: No

---

## [Editor Report · Acceptance letter]

PCOMPBIOL-D-25-00615R1

Personalizing chemotherapy drug selection using a novel transcriptomic chemogram

Dear Dr Scott,

I am pleased to inform you that your manuscript has been formally accepted for publication in PLOS Computational Biology. Your manuscript is now with our production department and you will be notified of the publication date in due course.

With kind regards,

Zsofia Freund
